# A two-part model to estimate inpatient, outpatient, prescribing and care home costs associated with atrial fibrillation in Scotland

Giorgio Ciminata ,[1] Claudia Geue,[1] Peter Langhorne,[2] Olivia Wu[1]

¹Health Economics and Health Technology Assessment (HEHTA), University of Glasgow, Glasgow, UK
²Institute of Cardiovascular and Medical Sciences, University of Glasgow, Glasgow, UK

**Correspondence to**
Mr Giorgio Ciminata;
g.ciminata.1@research.gla.ac.uk

## ABSTRACT

**Objective** This study aimed to estimate global inpatient, outpatient, prescribing and care home costs for patients with atrial fibrillation using population-based, individual-level linked data.

**Design** A two-part model was employed to estimate the probability of resource utilisation and costs conditional on positive utilisation using individual-level linked data.

**Settings** Scotland, 5 years following first hospitalisation for AF between 1997 and 2015.

**Participants** Patients hospitalised with a known diagnosis of AF or atrial flutter.

**Primary and secondary outcome measures** Inpatient, outpatient, prescribing and care home costs.

**Results** The mean annual cost for a patient with AF was estimated at £3785 (95% CI £3767 to £3804). Inpatient admissions and outpatient visits accounted for 79% and 8% of total costs, respectively; prescriptions and care home stay accounted for 7% and 6% of total costs. Inpatient cost was the main driver across all age groups. While inpatient cost contributions (~80%) were constant between 0 and 84 years, they decreased for patients over 85 years. This is offset by increasing care home cost contributions. Mean annual costs associated with AF increased significantly with increasing number of comorbidities.

**Conclusion** This study used a contemporary and representative cohort, and a comprehensive approach to estimate global costs associated with AF, taking into account resource utilisation beyond hospital care. While overall costs, considerably affected by comorbidity, did not increase with increasing age, care home costs increased proportionally with age. Inpatient admission was the main contributor to the overall financial burden of AF, highlighting the need for improved mechanisms of early diagnosis to prevent hospitalisations.

## Strengths and limitations of this study

► Costs are estimated through an incidence-based approach using patient-level morbidity records.
► Sufficient follow-up time is used to capture all relevant global costs to generate a contemporary estimate of health and care home costs related to atrial fibrillation (AF).
► Scotland offers a robust record linkage system, where administrative patient-level health data are routinely collected.
► Data on primary care consultations were not available for linkage at a national level; however, the impact this might have on overall costs is expected to be small.
► The potential risk of AF going undiagnosed and clinical miscoding of morbidity records may lead to an underestimation of the AF cohort and associated costs.

## INTRODUCTION

Atrial fibrillation (AF) is the most common form of arrhythmia. In Scotland, AF affects 1.8% of the adult population, and rises to 6% among those aged 65 years or over.[1] In an ageing population, AF has a substantial impact on the economic burden of the healthcare system.

A number of cost analyses on estimating the economic burden of AF exist. The majority of these studies used various definition of the AF study population, based on data sourced from administrative database,[2–4] health insurance databases,[2 5–7] hospital records[8 9] and surveys.[10] Direct medical costs related to inpatient admissions, outpatient visits, as well as prescriptions have been included in these estimates[2–10]; indirect costs related to loss of productivity have been estimated among patients who were at working ages.[6 7]

There is a lack of generalisable studies based on large national population datasets that examine the total and the distribution of costs associated with AF.[11] The aim of this study was to quantify the inpatient, outpatient, prescribing and care home costs associated with AF over a 5-year period. Using record linkage of national datasets from Scotland, we also examined the distribution of costs that are attributable to AF.

## METHODS

Cost analyses or cost of illness studies typically adopt either the prevalence-based or incidence-based approaches.[12] In the context of AF, the prevalence-based approach determines costs attributable to all cases of AF in a given year, while the incidence-based approach determines costs of new cases of AF in a given time period. In the present study, costs were estimated with an incidence-based approach. A further distinction between costing analyses is between the medicalised and the global comprehensive approaches.

In the first case, only expenditures directly attributable to a particular disease are used for estimating the overall costs. While the medicalised approach can be used to identify highly specific expenditures, it may also lead to underestimation or overestimation of the economic burden of a given disease; this may happen when cost estimation is not adequately adjusted for confounders highly correlated with the disease of interest. Conversely, the global comprehensive approach, used in this analysis, includes all the expenditures incurred by a population with a particular disease.[13] These expenditures are not necessarily related to the disease of interest; for instance, expenditures related to orthopaedics surgery or cancer treatment incurred by a patient with AF will count towards the global comprehensive cost of AF.

## DATA

Data were obtained from the Information Services Division (ISD) of NHS Scotland as part of a wider project that used routinely collected data to evaluate clinical effectiveness and cost-effectiveness of direct oral anticoagulants in the prevention of stroke in the AF population. Inpatient records for patients with a diagnosis of AF or atrial flutter between 1997 and 2015 were extracted from the General Acute Inpatient and Day Case Scottish Morbidity Records 01 (SMR01). These records contain all general acute admissions, categorised as inpatients or day cases, discharged from non-obstetric and non-psychiatric specialties.[14] Incident AF events (ICD10 code I48) were identified using all six diagnostic positions in SMR01, with a look back period of 5 years to minimise double counting. After checking for data entry errors and removal of duplicate records, the final AF cohort consisting of 278 286 individuals hospitalised with a diagnosis of AF or atrial flutter was identified.

Individual-level data linkage was then carried out with outpatient clinic attendance (Outpatient Attendance Scottish Morbidity Records 00; SMR00), the prescribing information system (PIS), care home census and mortality records (National Records for Scotland, NRS). Records from SMR00 include information on new and follow-up outpatient appointments for any clinical specialty.[15] The PIS database includes prescribing records for all medicines and their associated costs, which are prescribed and dispensed by community pharmacies, dispensing doctors and a small number of specialist appliance suppliers.[16]

| Table 1 | Baseline characteristics of patients with AF |
| --- | --- |
| **Characteristics** | **N (%)** |
| Number of patients | 278 286 |
| Mean age at first admission* (SD)** (range) | 74* (12.5)** (0–108) |
| Sex | |
| Male | 139 928 (50.3) |
| Female | 138 358 (49.7) |
| Health boards | |
| Greater Glasgow and Clyde | 61 822 (22.2) |
| Lothian | 41 169 (14.8) |
| Lanarkshire | 31 049 (11.2) |
| Grampian | 25 728 (9.3) |
| Ayrshire and Arran | 22 003 (7.9) |
| Tayside | 25 003 (9.0) |
| Fife | 17 954 (6.5) |
| Highland | 18 929 (6.9) |
| Forth Valley | 13 664 (4.9) |
| Dumfries and Galloway | 9798 (3.5) |
| Borders | 7222 (2.6) |
| Western Isles | 1868 (0.7) |
| Shetland | 1036 (0.4) |
| Orkney | 1041 (0.4) |
| Geography | |
| Large/urban | 106 868 (38.4) |
| Other/urban | 82 601 (29.7) |
| Accessible small towns | 24 938 (9.0) |
| Remote small towns | 8272 (3.0) |
| Very remote small towns | 3828 (1.4) |
| Accessible rural | 30 826 (11.1) |
| Remote rural | 10 371 (3.7) |
| Very remote rural | 10 087 (3.6) |
| SIMD quintile | |
| 1 | 62 730 (22.5) |
| 2 | 62 632 (22.5) |
| 3 | 55 943 (20.1) |
| 4 | 50 691 (18.2) |
| 5 | 46 279 (16.6) |
| Comorbidity | |
| No comorbidity | 40 502 (14.6) |
| 1 comorbidity | 53 651 (19.3) |
| >1 comorbidities | 184 133 (66.2) |
| Re-hospitalised (any condition) | 179 494 (64.5) |
| Admitted to care home | 7235 (2.6) |
| Mortality | |
| Alive | 204 690 (73.6) |
| Dead | 73 596 (26.4) |

SIMD, Scottish Index of Multiple Deprivation.

**Table 2** Regression results: probability of healthcare resources utilisation and cost estimation

| Covariates | Probability (first modelling part) | | Cost ratios (second modelling part) | |
|---|---|---|---|---|
| | Coefficient (95% CI) | SE | Coefficient (95% CI) | SE |
| Age group (years) | | | | |
| 0–49 | Reference | | | |
| 50–54 | 0.329 (0.260 to 0.398) | 0.035 | 0.036 (−0.016 to 0.087) | 0.026 |
| 55–59 | 0.388 (0.326 to 0.450) | 0.031 | 0.081 (0.036 to 0.127) | 0.023 |
| 60–64 | 0.464 (0.407 to 0.521) | 0.029 | 0.124 (0.082 to 0.166) | 0.021 |
| 65–69 | 0.486 (0.432 to 0.540) | 0.028 | 0.157 (0.116 to 0.198) | 0.021 |
| 70–74 | 0.479 (0.426 to 0.533) | 0.027 | 0.213 (0.174 to 0.252) | 0.020 |
| 75–79 | 0.536 (0.482 to 0.590) | 0.027 | 0.222 (0.183 to 0.260) | 0.020 |
| 80–84 | 0.431 (0.375 to 0.486) | 0.028 | 0.286 (0.246 to 0.326) | 0.020 |
| 85–89 | 0.378 (0.318 to 0.437) | 0.030 | 0.375 (0.332 to 0.417) | 0.021 |
| 90−max | 0.150 (0.083 to 0.217) | 0.034 | 0.516 (0.468 to 0.564) | 0.025 |
| Sex | | | | |
| Male | Reference | | | |
| Female | 0.045 (0.028 to 0.062) | 0.009 | 0.054 (0.044 to 0.064) | 0.005 |
| Date of admission | 0.169 (0.167 to 0.171) | 0.001 | −0.024 (−0.025 to −0.023) | 0.001 |
| SIMD quintile | | | | |
| 1 | Reference | | | |
| 2 | 0.027 (−0.018 to 0.071) | 0.023 | −0.055 (−0.080 to −0.031) | 0.012 |
| 3 | −0.041 (−0.086 to 0.003) | 0.023 | −0.080 (−0.106 to −0.054) | 0.013 |
| 4 | −0.046 (−0.091 to −0.002) | 0.023 | −0.116 (−0.141 to −0.090) | 0.013 |
| 5 | −0.072 (−0.117 to −0.027) | 0.023 | −0.147 (−0.172 to −0.122) | 0.013 |
| Geography | | | | |
| Large urban | Reference | | | |
| Other urban | −0.130 (−0.156 to −0.105) | 0.013 | −0.023 (−0.037 to −0.009) | 0.007 |
| Accessible small towns | −0.153 (−0.187 to −0.119) | 0.017 | −0.041 (−0.060 to −0.022) | 0.010 |
| Accessible rural | −0.197 (−0.230 to −0.165) | 0.016 | −0.043 (−0.062 to −0.024) | 0.010 |
| Remote small towns | −0.145 (−0.197 to −0.093) | 0.027 | 0.009 (−0.023 to 0.041) | 0.016 |
| Remote rural | −0.288 (−0.335 to −0.241) | 0.024 | −0.036 (−0.065 to −0.007) | 0.015 |
| Very remote small towns | −0.380 (−0.459 to −0.300) | 0.041 | −0.057 (−0.107 to −0.006) | 0.026 |
| Very remote rural | −0.346 (−0.407 to −0.284) | 0.031 | −0.061 (−0.102 to −0.020) | 0.021 |
| Health boards | | | | |
| Great Glasgow and Clyde | Reference | | | |
| Lothian | −0.044 (−0.075 to −0.014) | 0.016 | −0.033 (−0.049 to −0.017) | 0.008 |
| Lanarkshire | −0.005 (−0.038 to 0.029) | 0.017 | −0.063 (−0.081 to −0.045) | 0.009 |
| Ayrshire and Arran | −0.358 (−0.394 to −0.321) | 0.019 | −0.046 (−0.068 to −0.024) | 0.011 |
| Grampian | 0.017 (−0.019 to 0.054) | 0.019 | −0.059 (−0.078 to −0.039) | 0.010 |
| Tayside | −0.402 (−0.436 to −0.368) | 0.018 | −0.083 (−0.103 to −0.062) | 0.010 |
| Fife | −0.059 (−0.101 to −0.017) | 0.022 | −0.009 (−0.033 to 0.016) | 0.012 |
| Highland | −0.175 (−0.225 to −0.124) | 0.026 | −0.046 (−0.077 to −0.015) | 0.016 |
| Forth Valley | −0.477 (−0.518 to −0.436) | 0.021 | −0.109 (−0.135 to −0.082) | 0.013 |
| Dumfries and Galloway | −0.303 (−0.352 to −0.253) | 0.025 | −0.134 (−0.164 to −0.104) | 0.015 |
| Borders | −0.501 (−0.554 to −0.449) | 0.027 | −0.086 (−0.120 to −0.052) | 0.017 |
| Western Isles | −1.072 (−1.171 to −0.974) | 0.050 | 0.457 (0.381 to 0.533) | 0.039 |
| Orkney | −0.362 (−0.492 to −0.232) | 0.066 | −0.029 (−0.117 to 0.059) | 0.045 |
| Shetland | −0.495 (−0.622 to −0.368) | 0.065 | −0.076 (−0.171 to 0.018) | 0.048 |

Continued

**Table 2** Continued

| Covariates | Probability (first modelling part) | | Cost ratios (second modelling part) | |
| --- | --- | --- | --- | --- |
| | Coefficient (95% CI) | SE | Coefficient (95% CI) | SE |
| Mortality within 5 years | | | | |
| Alive | Reference | | | |
| Dead | 0.418 (0.376 to 0.461) | 0.022 | 0.652 (0.630 to 0.674) | 0.011 |
| Comorbidity | | | | |
| No comorbidities | Reference | | | |
| 1 comorbidity | 0.666 (0.567 to 0.766) | 0.051 | 0.374 (0.299 to 0.450) | 0.038 |
| >1 comorbidities | 1.205 (1.021 to 1.390) | 0.094 | 0.990 (0.910 to 1.070) | 0.041 |

SIMD, Scottish Index of Multiple Deprivation.

The quality of PIS data is guaranteed by an electronic data capture, and it passes several stages of quality control before and after data are submitted.[17] The care home census combines the former Residential Care Home Census (run by the Scottish Government) and the Private Nursing Homes Census (run by ISD Scotland). Items reported in the care home census include discharge dates to care home residency such as NHS and private nursing homes, as well as an indication on whether nursing care is required.[16]

Patients were followed up for 5 years following incident AF event in terms of their healthcare resource use, care home admissions and mortality. Since AF is often a precursor of stroke and cardiovascular conditions, an estimation of costs for a period of 5 years post AF event would allow us to fully capture costs associated with a patient with AF.

## COSTING
Inpatient care costs were obtained from the latest (2013/2014) Scottish National Tariff (SNT), a list of standard average prices based on Healthcare Resource Groups (HRGs).[17 18] The SNT uses HRG4 for grouping clinically similar treatments that use similar levels of healthcare resources. After defining a total cost per episode, the total cost for a continuous inpatient stay (CIS) was calculated.

A CIS describes the entire duration of an inpatient stay from the date of admission to the date of discharge and can consist of several episodes in different specialties. Since the SNT is based on spells of care (inpatient stay within the same specialty) rather than individual inpatient episodes or a CIS, a CIS was partitioned into spells when a change in specialty occurred.[17] If within a CIS, two or more episodes were in the same specialty, only the highest incurred cost was taken into account, and the remaining episodes were replaced with a zero cost. Outpatient costs were obtained by assigning outpatient specialty costs to outpatient attendances.[17] Unit costs were specific to whether the outpatient attendance took place at a consultant-led or nurse-led clinic.[15]

The cost of each prescription dispensed per patient was obtained from PIS.[19] First, the price per unit was obtained by dividing the item price by the pack size. Second, the total number of items dispensed was obtained by multiplying the number of items dispensed by the number of instalments. Care home costs, obtained from the care home census, were based on length of stay or residency. Care home residency was established from care home census records, reporting admission to a care home-like structure.[16] An average of care home charges for long stay residents was calculated using information on whether nursing care was provided or not. The average weekly care home charge was expressed per day, so that only the effective days spent in a care home were costed. The tariffs used for costing account for inflation, therefore further cost adjustment was not needed.

## ECONOMETRIC MODEL
Healthcare expenditure data are typically characterised by (1) a significant proportion of zero-cost observations for individuals who have not used any healthcare resources in a given time period, and (2) a skewed distribution for positive costs. A two-part model was used.[20 21]

In the first part of the model, the probability of using a healthcare service in a given time period was estimated using a probit model (online supplementary equation I). The same explanatory variables were used in the second part of the model, with a gamma distribution and log link, estimating costs conditional on having incurred positive costs (online supplementary equation II). Mean costs per patient per year following their incident AF event were calculated by multiplying first and second modelling parts (online supplementary equation III).

In order to account for the skewed nature of cost data, generalised linear models (GLMs) were used. These were compared against ordinary least squares regression (OLS) and log-transformed OLS by means of the Akaike information criterion (AIC), which measures goodness of fit. When comparing the different models, GLM reported the lowest AIC, indicating the best fit for the given set of

**Table 3** Regression results: probability of healthcare resources utilisation and cost estimation (alive at the end of the 5-year follow-up period)

| Covariates | Probability (first modelling part) | | Cost ratios (second modelling part) | |
| --- | --- | --- | --- | --- |
| | Coefficient (95% CI) | SE | Coefficient (95% CI) | SE |
| Age group (years) | | | | |
| 0–49 | Reference | | | |
| 50–54 | 0.352 (0.282 to 0.422) | 0.036 | 0.067 (0.013 to 0.120) | 0.027 |
| 55–59 | 0.424 (0.361 to 0.488) | 0.032 | 0.148 (0.098 to 0.199) | 0.026 |
| 60–64 | 0.528 (0.470 to 0.586) | 0.030 | 0.218 (0.174 to 0.263) | 0.023 |
| 65–69 | 0.571 (0.516 to 0.627) | 0.028 | 0.292 (0.248 to 0.336) | 0.022 |
| 70–74 | 0.603 (0.549 to 0.658) | 0.028 | 0.412 (0.371 to 0.454) | 0.021 |
| 75–79 | 0.684 (0.630 to 0.739) | 0.028 | 0.484 (0.443 to 0.525) | 0.021 |
| 80–84 | 0.572 (0.516 to 0.628) | 0.028 | 0.615 (0.572 to 0.659) | 0.022 |
| 85–89 | 0.496 (0.435 to 0.557) | 0.031 | 0.805 (0.756 to 0.854) | 0.025 |
| 90—max | 0.206 (0.134 to 0.279) | 0.037 | 1.044 (0.981 to 1.106) | 0.032 |
| Sex | | | | |
| Male | Reference | | | |
| Female | 0.067 (0.048 to 0.086) | 0.010 | 0.050 (0.037 to 0.063) | 0.007 |
| Date of admission | 0.171 (0.170 to 0.173) | 0.001 | −0.059 (−0.060 to −0.057) | 0.001 |
| SIMD quintile | | | | |
| 1 | Reference | | | |
| 2 | 0.021 (−0.009 to 0.050) | 0.015 | −0.052 (−0.071 to −0.033) | 0.010 |
| 3 | −0.023 (−0.054 to 0.008) | 0.016 | −0.081 (−0.101 to −0.060) | 0.011 |
| 4 | −0.045 (−0.077 to −0.014) | 0.016 | −0.117 (−0.138 to −0.096) | 0.011 |
| 5 | −0.051 (−0.083 to −0.020) | 0.016 | −0.160 (−0.181 to −0.139) | 0.011 |
| Geography | | | | |
| Large urban | Reference | | | |
| Other urban | −0.140 (−0.169 to −0.112) | 0.014 | −0.030 (−0.049 to −0.012) | 0.010 |
| Accessible small towns | −0.172 (−0.210 to −0.134) | 0.019 | −0.052 (−0.077 to −0.026) | 0.013 |
| Accessible rural | −0.217 (−0.253 to −0.181) | 0.018 | −0.061 (−0.086 to −0.037) | 0.013 |
| Remote small towns | −0.145 (−0.203 to −0.087) | 0.030 | −0.007 (−0.048 to 0.035) | 0.021 |
| Remote rural | −0.319 (−0.371 to −0.268) | 0.026 | −0.064 (−0.101 to −0.027) | 0.019 |
| Very remote small towns | −0.404 (−0.491 to −0.318) | 0.044 | −0.098 (−0.161 to −0.036) | 0.032 |
| Very remote rural | −0.360 (−0.428 to −0.293) | 0.034 | −0.087 (−0.138 to −0.035) | 0.026 |
| Health boards | | | | |
| Great Glasgow and Clyde | Reference | | | |
| Lothian | −0.055 (−0.090 to −0.020) | 0.018 | −0.051 (−0.072 to −0.030) | 0.011 |
| Lanarkshire | 0.003 (−0.034 to 0.040) | 0.019 | −0.072 (−0.095 to −0.048) | 0.012 |
| Ayrshire and Arran | −0.396 (−0.436 to −0.355) | 0.021 | −0.064 (−0.093 to −0.035) | 0.015 |
| Grampian | 0.029 (-0.013 to 0.070) | 0.021 | −0.051 (−0.077 to −0.026) | 0.013 |
| Tayside | −0.453 (−0.491 to −0.415) | 0.019 | −0.094 (−0.120 to −0.067) | 0.014 |
| Fife | −0.087 (−0.134 to −0.040) | 0.024 | −0.024 (−0.057 to 0.008) | 0.017 |
| Highland | −0.191 (−0.247 to −0.135) | 0.029 | −0.037 (−0.075 to 0.001) | 0.020 |
| Forth Valley | −0.520 (−0.566 to −0.474) | 0.023 | −0.108 (−0.141 to −0.074) | 0.017 |
| Dumfries and Galloway | −0.314 (−0.369 to −0.259) | 0.028 | −0.166 (−0.206 to −0.127) | 0.020 |
| Borders | −0.547 (−0.605 to −0.489) | 0.030 | −0.099 (−0.144 to −0.054) | 0.023 |
| Western Isles | −1.164 (−1.264 to −1.063) | 0.051 | 0.139 (0.057 to 0.221) | 0.042 |
| Orkney | −0.394 (−0.535 to −0.252) | 0.072 | 0.002 (−0.114 to 0.117) | 0.059 |
| Shetland | −0.605 (−0.740 to −0.470) | 0.069 | −0.044 (−0.172 to 0.085) | 0.066 |

**Table 3** Continued

| Covariates | Probability (first modelling part) | | Cost ratios (second modelling part) | |
|---|---|---|---|---|
| | Coefficient (95% CI) | SE | Coefficient (95% CI) | SE |
| Comorbidity | | | | |
| No comorbidities | Reference | | | |
| 1 comorbidity | 0.705 (0.602 to 0.808) | 0.052 | 0.432 (0.352 to 0.513) | 0.041 |
| >1 comorbidities | 1.165 (0.974 to 1.357) | 0.098 | 1.133 (1.041 to 1.226) | 0.047 |

SIMD, Scottish Index of Multiple Deprivation.

data. A user-written STATA program 'glmdiagnostic.do',[20] performing four different tests simultaneously, was used to identify the most appropriate distributional family and link function.

## Econometric model covariates

The two-part model adjusted for age, sex, year of inpatient admission, socioeconomic status, urban–rural classification, health board, comorbidities and mortality. These covariates are considered to be the main confounders that have an effect on costs incurred by an AF population. We controlled for age because AF and associated comorbidities are age-related conditions and may have an impact on the overall costs. We also assumed costs to vary between men and women, in particular those for care home residency. Variation in healthcare utilisation and associated costs and care home residency by socioeconomic status is controlled for using the Scottish Index of Multiple Deprivation (SIMD).

The SIMD reflects areas of multiple deprivation ranked from the most to the least deprived and expressed as quintiles where the most and the least deprived areas are represented by 1 and 5, respectively.[22] In Scotland, there are 14 regional health boards responsible for the provision of healthcare.[23] Hence, potential differences in healthcare utilisation and prescribing costs may reflect variation in clinical practice and prescribing behaviour rather than the ability of patients to access care. Patients living in urban areas may have easier access to care compared with patients living in more remote areas, which is controlled for including the eightfold classification measuring rurality.[24]

Patients with one or more comorbidities are expected to incur significantly higher costs than those with none. We accounted for this by including the Charlson Comorbidity Index, where 1 indicates the absence of comorbidities, 2 the presence of only a single comorbidity and 3 the presence of more than one comorbidity.[25] Two interaction terms between age and comorbidities, and mortality and SIMD were included in the econometric model. Intuitively, a relationship of direct proportionality between age and comorbidities suggests that the level of comorbidities increases as patients get older. Similarly, the socioeconomic status may significantly influence the rate of socioeconomic inequalities in mortality.[26]

## SENSITIVITY ANALYSES

In order to ascertain whether mortality had an impact on overall AF-related healthcare costs, average annual cost per patient by age and for each health or care home sector was estimated for patients who were alive and those who were dead at the end of the 5-year follow-up period. The two econometric models (Equation IV and V, please see online online supplementary equation IV and V) followed the same structure of the model described in the previous section and used for the main analysis; however, those models were not adjusted for mortality.

## Patients and public involvement

There was no patients or public involvement.

## RESULTS
### Cohort characteristics

Of the 278 286 patients with AF with a mean age of 74 years (SD 12.5), the majority were identified in the two largest urban health board areas (Greater Glasgow and Clyde and Lothian), accounting for 22.2% and 14.8%, respectively. This is also reflected in our categorisation of geographical areas, where large urban represented 38.4% and other urban areas represented 29.7% of the total AF cohort. Greater proportion of patients live in areas belonging to the most deprived quintile compared with those living in the least deprived areas—SIMD quintile 1 and quintile 5 representing 22.5% and 16.6% of the AF cohort, respectively (table 1).

### Econometric modelling results

Regression results for both modelling parts are presented in table 2. Overall, an inversely U-shaped association between age and the likelihood of utilising any health or social care services was observed—a gradual increment in the likelihood in resource use with advancing age up to 80 years, when compared with the reference group (0–49 years), while patients 80 years or older showing a decreased probability of utilising healthcare services. However, this association was not observed in the second modelling part model, estimating costs conditional on having incurred positive costs, where a statistically significant gradient between age and costs indicated increasing costs as the cohort ages.

 Ciminata G, *et al. BMJ Open* 2020;**10**:e028575. doi:10.1136/bmjopen-2018-028575

**Table 4** Regression results: probability of healthcare resources utilisation and cost estimation (dead at the end of the 5-year follow-up period)

| Covariates | Probability (first modelling part) Coefficient (95% CI) | SE | Cost ratios (second modelling part) Coefficient (95% CI) | SE |
|---|---|---|---|---|
| **Age group (years)** | | | | |
| 0–49 | Reference | | | |
| 50–54 | 0.150 (−0.125 to 0.426) | 0.141 | −0.112 (−0.405 to 0.180) | 0.149 |
| 55–59 | 0.134 (−0.098 to 0.366) | 0.118 | −0.093 (−0.334 to 0.147) | 0.123 |
| 60–64 | 0.129 (−0.080 to 0.338) | 0.107 | 0.000 (−0.208 to 0.209) | 0.106 |
| 65–69 | 0.129 (−0.067 to 0.326) | 0.101 | −0.011 (−0.212 to 0.189) | 0.102 |
| 70–74 | 0.107 (−0.084 to 0.298) | 0.097 | 0.016 (−0.180 to 0.213) | 0.100 |
| 75–79 | 0.128 (−0.059 to 0.315) | 0.095 | −0.005 (−0.198 to 0.189) | 0.099 |
| 80–84 | 0.132 (−0.053 to 0.318) | 0.095 | 0.056 (−0.136 to 0.247) | 0.098 |
| 85–89 | −0.048 (−0.233 to 0.137) | 0.094 | 0.066 (−0.126 to 0.257) | 0.098 |
| 90—max | −0.518 (−0.702 to −0.333) | 0.094 | 0.097 (−0.095 to 0.290) | 0.098 |
| **Sex** | | | | |
| Male | Reference | | | |
| Female | 0.048 (0.033 to 0.063) | 0.008 | 0.028 (0.014 to 0.043) | 0.007 |
| **Date of admission** | −0.040 (−0.042 to −0.039) | 0.001 | 0.004 (0.002 to 0.005) | 0.001 |
| **SIMD quintile** | | | | |
| 1 | Reference | | | |
| 2 | 0.033 (0.011 to 0.055) | 0.011 | 0.015 (−0.005 to 0.036) | 0.011 |
| 3 | 0.058 (0.034 to 0.082) | 0.012 | −0.008 (−0.030 to 0.015) | 0.012 |
| 4 | 0.065 (0.039 to 0.090) | 0.013 | −0.017 (−0.041 to 0.007) | 0.012 |
| 5 | 0.113 (0.088 to 0.138) | 0.013 | −0.024 (−0.049 to 0.000) | 0.012 |
| **Geography** | | | | |
| Large urban | Reference | | | |
| Other urban | −0.010 (−0.032 to 0.012) | 0.011 | −0.033 (−0.054 to −0.012) | 0.011 |
| Accessible small towns | −0.006 (−0.036 to 0.025) | 0.015 | −0.049 (−0.077 to −0.021) | 0.014 |
| Accessible rural | −0.031 (−0.060 to −0.001) | 0.015 | −0.036 (−0.064 to −0.008) | 0.014 |
| Remote small towns | −0.054 (−0.102 to −0.005) | 0.025 | 0.003 (−0.042 to 0.049) | 0.023 |
| Remote rural | −0.038 (−0.084 to 0.009) | 0.024 | −0.012 (−0.057 to 0.034) | 0.023 |
| Very remote small towns | −0.065 (−0.147 to 0.017) | 0.042 | 0.036 (−0.052 to 0.123) | 0.045 |
| Very remote rural | 0.014 (−0.051 to 0.078) | 0.033 | −0.002 (−0.068 to 0.065) | 0.034 |
| **Health boards** | | | | |
| Great Glasgow and Clyde | Reference | | | |
| Lothian | 0.029 (0.004 to 0.055) | 0.013 | 0.029 (0.006 to 0.053) | 0.012 |
| Lanarkshire | −0.052 (−0.080 to −0.023) | 0.014 | −0.034 (−0.061 to −0.008) | 0.013 |
| Ayrshire and Arran | −0.122 (−0.155 to −0.089) | 0.017 | 0.011 (−0.020 to 0.042) | 0.016 |
| Grampian | 0.075 (0.044 to 0.106) | 0.016 | −0.057 (−0.086 to −0.028) | 0.015 |
| Tayside | −0.024 (−0.056 to 0.007) | 0.016 | −0.061 (−0.089 to −0.033) | 0.014 |
| Fife | −0.028 (−0.064 to 0.008) | 0.018 | 0.047 (0.012 to 0.082) | 0.018 |
| Highland | 0.034 (−0.015 to 0.084) | 0.025 | −0.065 (−0.117 to −0.013) | 0.027 |
| Forth Valley | −0.060 (−0.099 to −0.021) | 0.020 | −0.123 (−0.161 to −0.085) | 0.019 |
| Dumfries and Galloway | −0.027 (−0.074 to 0.020) | 0.024 | −0.014 (−0.058 to 0.029) | 0.022 |
| Borders | −0.058 (−0.112 to −0.005) | 0.027 | −0.023 (−0.074 to 0.029) | 0.026 |

**Table 4** Continued

| Covariates | Probability (first modelling part) | | Cost ratios (second modelling part) | |
|---|---|---|---|---|
| | Coefficient (95% CI) | SE | Coefficient (95% CI) | SE |
| Western Isles | −0.033 (−1.168 to 1.102) | 0.579 | 0.305 (−0.165 to 0.775) | 0.240 |
| Orkney | 0.191 (0.055 to 0.327) | 0.069 | −0.180 (−0.317 to −0.042) | 0.070 |
| Shetland | −0.031 (−0.170 to 0.108) | 0.071 | −0.187 (−0.323 to −0.052) | 0.069 |
| Comorbidity | | | | |
| No comorbidities | Reference | | | |
| 1 comorbidity | −0.176 (−0.449 to 0.097) | 0.139 | 0.147 (−0.127 to 0.422) | 0.140 |
| >1 comorbidities | −0.256 (−0.491 to −0.021) | 0.120 | 0.626 (0.401 to 0.851) | 0.115 |

SIMD, Scottish Index of Multiple Deprivation.

The use of health or social care services and associated costs also increased significantly for patients living in the most deprived areas, when compared with patients living in areas with the lowest level of deprivation. The effect of socioeconomic status on healthcare utilisation was also measured for those who are alive at the end of the 5-year follow-up period through an interaction term between SIMD and mortality, but no statistically significant effect was found.

Full details of regression results for interaction terms are presented in the online supplementary table I.

For patients with comorbidities, the probabilities of utilising healthcare services were greater than the probability for those with no comorbidities. Although healthcare utilisation increased with the number of comorbidities, the interaction term between age and comorbidities indicated that as patients get older, the use of healthcare services on average is lower for patients with one or more comorbidities than those with none. The decrease in healthcare utilisation by age is more pronounced in patients with more comorbidities than in those with only one comorbidity. The difference in healthcare costs between comorbidity categories indicated that in the presence of one or more comorbidities, on average healthcare costs decrease as patients get older. Full details of regression results for patients who were alive and those who were dead at the end of the 5-year follow-up period are presented in tables 3 and 4, respectively, while regression results for interaction terms are presented in the online supplementary tables II and III.

## COST ESTIMATES

The estimated mean annual cost per AF patient was £3785 (95% CI £3767 to £3804). The estimated total costs and distribution of costs according to sex are shown in table 5.

While there is little difference between the total costs and the distribution of costs for inpatient, outpatient and prescription costs, the difference seems more pronounced when comparing the care home component

of costs (5% of total costs among male vs 7% of total costs among female).

The average annual cost per AF patient by age and for each health or care home sector is shown in figure 1. Considering the individual contribution of each cost component to the overall costs, inpatient cost was the main driver across all age groups. While inpatient cost contribution remained constant with an average contribution of about 80% to the overall costs for patients aged between 0 and 84 years, it decreased for patients over 85 years of age. Similar patterns were observed for outpatient and prescribing costs. On the contrary, the contribution of care home costs to the overall costs increased with age (0.5% for patients aged 0–49 years and approximately 11% for patients who are 90 years or older). The contribution of each setting to the total health and care home costs by the number of existing comorbidities is illustrated in figure 2. While inpatient and total costs vary

**Table 5** Average annual costs per patient hospitalised with AF by sex

| Sex | Cost estimates | |
|---|---|---|
| | Mean total cost (%) | 95% CI |
| Male | | |
| Inpatient | 2935 (79.99) | (2915 to 2955) |
| Outpatient | 31 (8.46) | (308 to 313) |
| Care home | 165 (4.50) | (154 to 177) |
| PIS | 242 (6.60) | (240 to 245) |
| Total | 3669 | (3872 to 3927) |
| Female | | |
| Inpatient | 3022 (77.49) | (3001 to 3042) |
| Outpatient | 310 (7.96) | (308 to 313) |
| Care home | 268 (6.88) | (255 to 281) |
| PIS | 259 (6.64) | (256 to 262) |
| Total | 3968 | (3872 to 3927) |

PIS, prescribing information system.

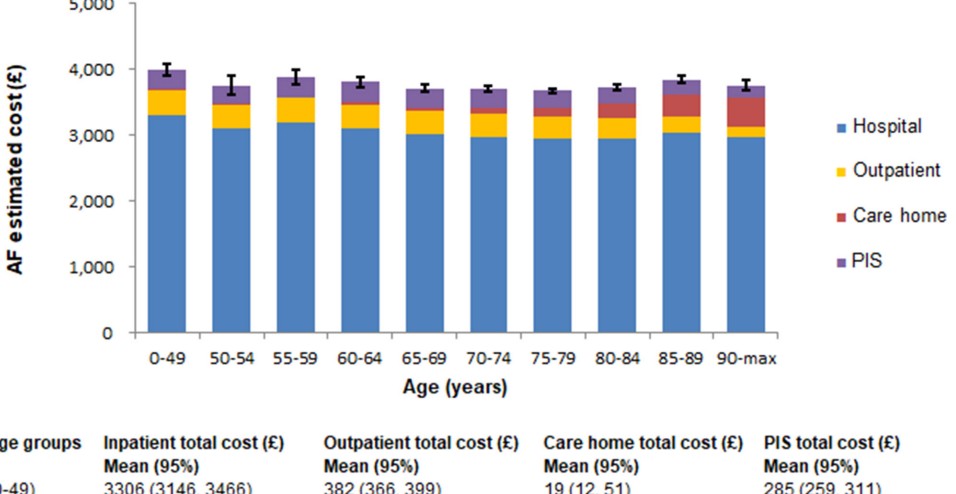

| Age groups | Inpatient total cost (£) Mean (95%) | Outpatient total cost (£) Mean (95%) | Care home total cost (£) Mean (95%) | PIS total cost (£) Mean (95%) |
|---|---|---|---|---|
| (0–49) | 3306 (3146, 3466) | 382 (366, 399) | 19 (12, 51) | 285 (259, 311) |
| (50–54) | 3107 (2979, 3236) | 367 (354, 379) | 19 (9, 47) | 270 (253, 287) |
| (55–59) | 3193 (3100, 3287) | 370 (361, 379) | 22 (2, 43) | 290 (278, 302) |
| (60–64) | 3107 (3044, 3170) | 364 (357, 371) | 47 (28, 67) | 293 (285, 302) |
| (65–69) | 3009 (2960, 3058) | 363 (358, 368) | 53 (37, 69) | 286 (280, 291) |
| (70–74) | 2969 (2931, 3007) | 355 (351, 359) | 95 (78, 111) | 282 (278, 287) |
| (75–79) | 2954 (2923, 2985) | 340 (336, 343) | 118 (104, 132) | 267 (264, 271) |
| (80–84) | 2962 (2930, 2994) | 299 (296, 303) | 225 (207, 242) | 247 (244, 250) |
| (85–89) | 3038 (3002, 3074) | 249 (245, 253) | 340 (318, 363) | 224 (221, 228) |
| (90–max) | 2971 (2926, 3017) | 161 (157, 165) | 435 (404, 466) | 191 (186, 195) |
| Total | 2978 (2964, 2993) | 310 (309, 312) | 226 (217, 235) | 250 (248, 252) |

**Figure 1** Average annual costs per patient hospitalised with AF by sector. Cost components with CI are presented for each age group. AF, atrial fibrillation; PIS, prescribing information system.

considerably with the number of comorbidities, outpatient and care home contributions remain fairly constant.

The estimated mean annual cost per AF patient alive at the end of the 5-year follow-up period was £3047 (95% CI £3027 to £3067). The average annual cost per AF patient by age and for each health or care home sector is presented in the online supplementary figure I. For these patients, inpatient cost was the main driver across

all age groups; a gradient between age and costs indicated increasing costs as the cohort ages. Similar patterns were observed for care home costs. On the contrary, outpatient and prescribing costs remained constant up to 74 years, but decreased slightly for older patients.

The estimated mean annual cost per AF patient who died during the 5-year follow-up period was £2304 (95% CI £2284 to £2324) (online supplementary figure II). For

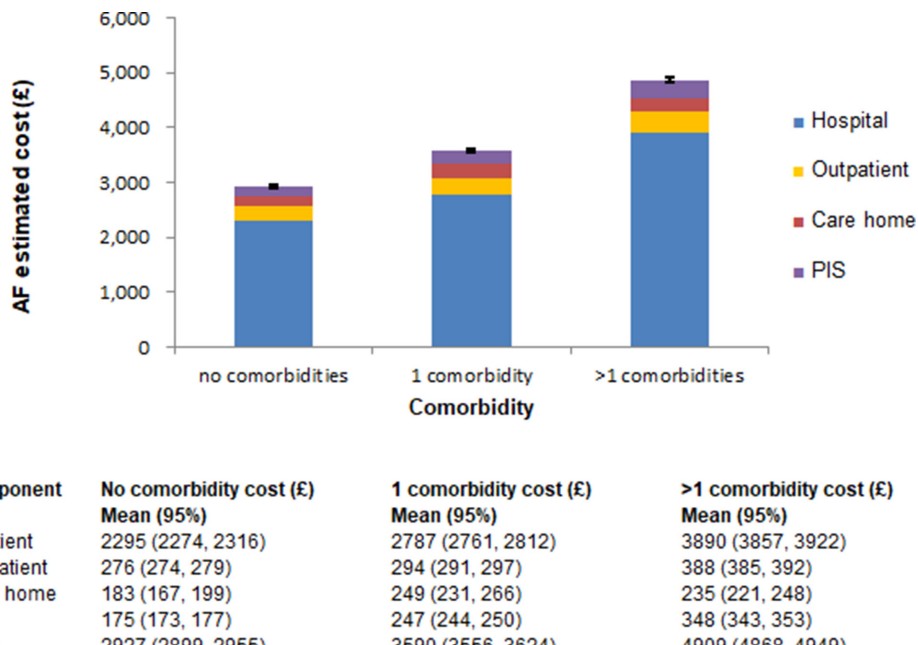

| Component | No comorbidity cost (£) Mean (95%) | 1 comorbidity cost (£) Mean (95%) | >1 comorbidity cost (£) Mean (95%) |
|---|---|---|---|
| Inpatient | 2295 (2274, 2316) | 2787 (2761, 2812) | 3890 (3857, 3922) |
| Outpatient | 276 (274, 279) | 294 (291, 297) | 388 (385, 392) |
| Care home | 183 (167, 199) | 249 (231, 266) | 235 (221, 248) |
| PIS | 175 (173, 177) | 247 (244, 250) | 348 (343, 353) |
| Total | 2927 (2899, 2955) | 3590 (3556, 3624) | 4909 (4868, 4949) |

**Figure 2** Average cost per patient hospitalised with AF by Charlson Comorbidity Index. Cost components with CI are presented for each comorbidity category. AF, atrial fibrillation; PIS, prescribing information system.

these patients, inpatient cost was the main driver across all age groups; a gradient between age and costs indicated decreasing costs as the cohort ages. This was also observed for outpatient and prescribing costs; but care home costs on average increased across age groups.

## DISCUSSION

A greater proportion of patients with AF were found in areas with the highest index of deprivation. This, combined with the likelihood for people living in the most deprived quintile having longer inpatient stays due to a lack of support at home, may explain the difference in inpatient care utilisation between patients from the most and the least deprived areas, with associated costs being higher for the former group. As AF is more likely to affect the elderly, so that costs were expected to increase with age. As health deteriorates with age, older age groups are assumed to make greater use of healthcare services, and therefore incur higher costs than younger age groups. However, age was found to have a modest impact on overall healthcare costs, being fairly consistent across age groups. This finding is in line with existing evidence indicating that healthcare expenditure depends not only on patients' calendar age but is also significantly associated with remaining lifetime.[27]

Any observed correlation between healthcare expenditure and age may therefore be attributable to the fact that the proportion of patients who are at the end of their lives is substantially greater in older rather than younger age groups.[27] On the other hand, comorbidity had a considerable effect on the overall costs, increasing significantly in patients with more than one comorbidity. However, the decrease in healthcare utilisation by age is more pronounced in patients with more comorbidities than in those with only one comorbidity.

Decreasing inpatient and outpatient costs for the oldest patients were offset by increasing care home costs, in particular for women. Indeed, the main cause for higher overall costs incurred by women is attributable to the higher likelihood for elderly women to reside in care homes.

Interestingly, care home contribution to the overall costs was noticeably lower for patients with multiple comorbidities than for those with none or one comorbidity. This may suggest that sicker patients are more likely to be in hospital than in a care home.

To date, only one single study published in 2004 has estimated the cost of AF in Scotland; the authors estimated the cost of AF in 1995/1996 with the medicalised approach, and projected these to the year 2000.[28] Previous work has focused on a 12-month follow-up, which seems limited in order to capture all healthcare resource utilisation for patients with AF. Our study offers a longer follow-up and a contemporary estimate of healthcare costs related to AF including all relevant care settings. Our study offers a distinct advantage over previous work as costs, rather than being based on extrapolated rates

using a prevalence-based approach,[28] are estimated with an incidence-based method using patient-level morbidity records. Using an incidence-based approach to costing and a broad perspective to capture the majority of costs associated with AF, several routinely collected administrative datasets from Scotland were combined, including care home utilisation.

Existing studies, including ours, regardless of econometric model choice and covariates used, show that costs due to inpatient admission are the main contributor to overall AF-related healthcare cost. This is a pertinent finding that may well support future policies on opportunistic screening in the population at risk of AF, and in particular in Scotland where one in three patients with AF are currently undiagnosed.[29]

The European AF management guidelines and the Scottish Cross-Party Group 'Heart Disease and Stroke' recently recommended that people who are 65 years or older and at risk of AF and associated comorbidities such as cardiovascular disease, diabetes or respiratory disease should be screened opportunistically in primary care, pharmacies or community settings.[29 30] With rigorous screening and appropriate treatment, hospitalisations could be avoided and costs reduced.

Although we have captured most healthcare sectors and related costs, we were not able to obtain national data on primary care consultations, as these data are currently not routinely available for linkage in Scotland. Not capturing these data may lead to an underestimation of the size of the AF cohort and associated costs. However, the cost associated with primary care consultations is expected to have a limited impact on the overall total AF-related costs. Such underestimation could also result from AF going undiagnosed and clinical miscoding of morbidity records. Nevertheless, by using a cohort of patients hospitalised with AF, we were able to capture more severe cases of AF. Prescribing and care home data were only available respectively from 2009 to 2012, their contribution to overall AF-related costs might also be underestimated. Other limitations are inherent to the nature of administrative data, such as missing records or incomplete data.

Further, we acknowledge the issue concerning attributing AF-related costs to patients with a structural heart disease, as AF may manifest subsequently because of this. In our analysis, we identified about 14% of AF patients with a structural heart disease; these were patients with systolic dysfunction, valvular heart disease or heart valve replacement. However, from the hospital data, it was not possible to establish causation between structural heart disease and AF.

In addition, this is likely to have a marginal impact on our conclusions, as the global comprehensive approach used in this study include expenditures that are not necessarily related to AF.

We also acknowledge that specifying whether patients had received cardiovascular procedures (eg, cardioversion, echocardiograms and angiograms) would improve the accuracy of our cost estimation, as it would indicate

whether costs should be attributable to AF or other forms of structural heart disease. However, this information is not currently available in our routinely collected data of hospital admissions.

Recognising these limitations, we were nevertheless able to harness high-quality patient-level linked data to identify a cohort of patients with AF and to estimate their associated healthcare utilisation and costs in Scotland.

The inclusion of all available cost components is crucial for establishing overall costs, as these often extend beyond hospitalisation. The study identifies hospitalisation as the main cost driver and suggests that the implementation of AF screening policies could substantially reduce AF-related healthcare costs. Most importantly, the study concludes that patient's age has a limited impact on the overall AF-related cost and therefore may contribute much less to future growth of AF-related cost in an ever-ageing Scottish population.

Future work will be able to use Scottish Stroke Care Audit (SSCA) records, allowing for the identification of additional patients with AF ; these are patients hospitalised with a stroke, where AF has been recorded in audit data as an underlying comorbidity.

Being able to complement inpatient records with SSCA records will allow us to capture more patients with AF in Scotland. Moreover, future research may be able to include indirect costs associated with productivity loss by linking morbidity and prescribing data to national data from the Department for Work and Pensions, for instance.

**Acknowledgements** We thank the members of the Information Services Division, National Services Scotland, for their support. We acknowledge the support from The Farr Institute @ Scotland. The Farr Institute @ Scotland is supported by a 10-funder consortium: Arthritis Research UK, the British Heart Foundation, Cancer Research UK, the Economic and Social Research Council, the Engineering and Physical Sciences Research Council, the Medical Research Council, the National Institute of Health Research, the National Institute for Social Care and Health Research (Welsh Assembly Government), the Chief Scientist Office (Scottish Government Health Directorates), the Wellcome Trust (MRC Grant No: MR/K007017/1).

**Contributors** GC, CG, PL and OW conceived the article. GC carried out the statistical analysis and prepared the first draft of the manuscript. GC, CG, PL and OW contributed to editing the manuscript and approved the final version submitted for publication.

**Funding** This work was supported by Farr Institute for Health Informatics Research, Scotland, MRC, grant number MR/K007017/1.

**Competing interests** None declared.

**Patient consent for publication** Not required.

**Ethics approval** No ethics approval was sought as this study does not require consenting/contacting patients directly. All data used by researchers are pseudonymised, data reported are aggregated to minimise risk of identification and output clearance is required.

**Provenance and peer review** Not commissioned; externally peer reviewed.

**Data availability statement** Data may be obtained from a third party and are not publicly available. All data underlying the analyses are confidential and subject to disclosure control. Data can only be obtained through application to Information Services Division (ISD) via the Public Benefit and Privacy Panel (PBPP).

**ORCID iD**
Giorgio Ciminata http://orcid.org/0000-0002-6905-6817

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
