## [Reviewer comments · BMJ Open]

ARTICLE DETAILS

TITLE (PROVISIONAL)	A Two-Part Model to estimate Inpatient, Outpatient, Prescribing and Care Home Costs associated with Atrial Fibrillation in Scotland
AUTHORS	Ciminata, Giorgio; Geue, Claudia; Langhorne, Peter; Wu, O

VERSION 1 – REVIEW

REVIEWER	Matthew Reynolds, MD, MSc Lahey Hospital & Medical Center, Burlington, MA, USA
REVIEW RETURNED	23-Jan-2019

GENERAL COMMENTS	1. The incident AF approach here is logical, but as the authors suggest in the Discussion, captures some but not all incident AF patients and would tend to identify “more severe cases.” In clinical practice, plenty of AF patients who have no symptoms or mild symptoms are diagnosed in ambulatory settings. By not capturing such patients, your study likely overestimates the annual costs of caring for AF patients at the patient level, though it may underestimate the total economic burden of the disease (which was not the aim of the study). This should be acknowledged 2. Was the cost of the index hospital admission for AF included for each patient, or only costs after that hospital admission? 3. The finding that overall costs did not increase substantially in the older age groups was surprising to me. How much did this have to do with mortality/censoring? The 1 or 5-year mortality rates for the AF patients across age strata are not shown, but might be of interest in helping to understand this. 4. It would be nice to proportions of patients utilizing different services over time. How many AF patients were re-hospitalized? How many wound up in “care-home”. I would be interested to see these proportions 5. While those concerned with reducing the economic burden of AF would be right to focus on hospital admissions, I am not sure, as suggested in a few places (e.g. p. 13) that increasing the detection of AF through earlier diagnosis will accomplish this. It has not been shown to. It is possible that detection of asymptomatic AF will allow for anticoagulation of at-risk patients, and this will reduce things like stroke. But, employing other therapies to “halt the progression” of asymptomatic AF, when detected, is detected is an unproven strategy that itself might incur increased treatment costs.
--

REVIEWER	Abdulla A Damluji Johns Hopkins University
REVIEW RETURNED	08-May-2019

GENERAL COMMENTS	In this manuscript Ciminata and colleagues aimed to quantify the inpatient, outpatient, prescribing and care home costs associated
--

with atrial fibrillation over a five-year period in Scotland. Further, they aimed to examine the distribution of costs attributed to atrial fibrillation utilizing a record-linkage of national datasets from Scotland. The paper is well written and presented. I have the following comments to the authors:

(1) In the introduction, the authors made the point that a number of analyses attempted to estimate the cost associated with AF. The note that these analyses are “selective cohorts”. Can the authors explain what they mean by that? My interpretation is that there is heterogeneity in the definition of the study population, but I am not sure if this is actually correct.

(2) The authors noted that there is a large range in the age of the study population in these studies (18 to >65 years of age) with AF mostly associated with structural heart disease, valvular heart disease, and other metabolic disorders seen in the later years. I am not sure where the cutoff of age 50 came from in the introduction section. I suggest that the authors avoid dichotomizing age, especially that there is less evidence that patients at a specific cutoff are prone to atrial arrhythmia than those below that cutoff. I agree with the authors though that older adults are at an increased risk for structural heart disease, systolic dysfunction, valvular heart disease and others, which in turn can manifest with AF (as a presenting disorder), but the alternate is true also. i.e. those who have heart failure for example, can progress with time in terms of their cardiomyopathy and result in AF...This makes the attribution of cost to AF challenging.

(3) A strong rationale on why the study population was decided to include patients who are older than 50 is needed. I feel that this cutoff is subjective specifically when looking at reference 11. If a comprehensive examination of cost attributed to AF is sought, all adults admitted with AF should be included. If this is an examination for older adults, who happen to be at the greatest risk for AF (with the least evidence given the systematic exclusion from clinical trials), a cutoff of > 75 years of age was suggested (Circulation 2016;133:2103–2122. DOI: 10.1161/CIR.0000000000000380)

(4) While AF can certainly be a “precursor” to stroke and cardiovascular disease, that is not truly the case everytime, specifically if we examine older adults with structural heart disease. For example, patients with mitral valve disease that progress into a severe form may result in a progressive dilation of the left atrial (and subsequent structural changes like fibrosis), and lead to atrial fibrillation. Attributing the cost of such patients to AF, rather to valvular heart disease is actually a major problem. In order to solve this problem, I think a sensitivity analysis to describe the proportion of patients with structural heart disease (at minimum systolic dysfunction, valvular heart disease, ...etc). The cost of AF in this setting should be presented in the context of other associated cardiovascular conditions.

(5) For costing, was that inflation adjusted? Please clarify...The description of costing is great!

(6) Page 6: typo (“skewed”) correct pls.

(7) Can the authors add another model that is conditional on survival (one year/five years)? (similar rationale to having conditioned on

	having positively incurred cost). Death is the biggest cost saver - I think it would be informative to estimate the cost attributed to AF among patients who are 5-year survivors. (8) Regarding covariates (i.e. adjustments), how about cardiovascular procedures (at minimum TEE/Cardioversions, holters, event monitors, echocardiograms, angiograms, PCIs, and other cardiovascular procedures?! (all associated with AF). Without adjusting for these, any incurred cost for a patient who was admitted to the hospital AF, and then received an angiogram/PCI will be attributed to AF, rather than coronary artery disease. (9) The discussion is nicely written, but I would expand it to include some of the points discussed above. Limitation sections should include the points above. (10) I would like to congratulate the authors on attempting to quantify the cost associated with AF in this heterogenous population. While I understand the complexity of the topic and effort that went into putting this work together (I enjoyed reading it), I feel that my overall concerns lie in clarifying the definition of the study population (why a cutoff of 50 was chosen rather than 65 or 75) and a better delineation of why cost was attributed to AF rather than other CV conditions or procedures. I am hopeful that the authors can address specially if this manuscript is considered for publication in a large platform like BMJ.
--	--

VERSION 1 – AUTHOR RESPONSE

Reviewer #1

1. The incident AF approach here is logical, but as the authors suggest in the Discussion, captures some but not all incident AF patients and would tend to identify “more severe cases.” In clinical practice, plenty of AF patients who have no symptoms or mild symptoms are diagnosed in ambulatory settings. By not capturing such patients, your study likely overestimates the annual costs of caring for AF patients at the patient level, though it may underestimate the total economic burden of the disease (which was not the aim of the study). This should be acknowledged

Authors’ response: The potential for underestimation of total economic burden of the disease is now acknowledged on page 15 of the main document.

2. Was the cost of the index hospital admission for AF included for each patient, or only costs after that hospital admission?

Authors’ response: Both costs were included into a single cost estimation (£3042 (£3027, £3057) presented in Figure 1.

3. The finding that overall costs did not increase substantially in the older age groups was surprising to me. How much did this have to do with mortality/censoring? The 1 or 5-year mortality rates for the AF patients across age strata are not shown, but might be of interest in helping to understand this.

Authors’ response: The reviewer raised an important point. We now include additional models for i) patients who survived and ii) those who died within the five years from AF index hospital admission to better distinguish overall costs. Regression results for these models are now presented in the Appendix and results are discussed in the main paper.

4. It would be nice to proportions of patients utilizing different services over time. How many AF patients were re-hospitalized? How many wound up in “care-home”. I would be interested to see these proportions

Authors’ response: Proportions of patients re-hospitalised or admitted to a care-home are now presented in Table 1. Proportions of patients who survived or died within the 5 years post AF incident event are also presented in Table 1.

5. While those concerned with reducing the economic burden of AF would be right to focus on hospital admissions, I am not sure, as suggested in a few places (e.g. p. 13) that increasing the detection of AF through earlier diagnosis will accomplish this. It has not been shown to. It is possible that detection of asymptomatic AF will allow for anticoagulation of at-risk patients, and this will reduce things like stroke. But, employing other therapies to “halt the progression” of asymptomatic AF, when detected, is detected is an unproven strategy that itself might incur increased treatment costs.

Authors’ response: A clearer explanation on policy implications is now provided on page 14 of the main document.

Reviewer # 2

In this manuscript, Ciminata and colleagues aimed to quantify the inpatient, outpatient, prescribing and care home costs associated with atrial fibrillation over a five-year period in Scotland. Further, they aimed to examine the distribution of costs attributed to atrial fibrillation utilizing a record-linkage of national datasets from Scotland. The paper is well written and presented. I have the following comments to the authors:

1. In the introduction, the authors made the point that a number of analyses attempted to estimate the cost associated with AF. The note that these analyses are “selective cohorts”. Can the authors explain what they mean by that? My interpretation is that there is heterogeneity in the definition of the study population, but I am not sure if this is actually correct.

Authors’ response: The interpretation is correct. A clearer sentence on this is now present on page 3 of the main document. The new sentence reads as: “The majority of these studies used various definition of the AF study population, based on data sourced from administrative database, health insurance databases, hospital records, and surveys”.

2. The authors noted that there is a large range in the age of the study population in these studies (18 to >65 years of age) with AF mostly associated with structural heart disease, valvular heart disease, and other metabolic disorders seen in the later years. I am not sure where the cutoff of age 50 came from in the introduction section. I suggest that the authors avoid dichotomizing age, especially that there is less evidence that patients at a specific cutoff are prone to atrial arrhythmia than those below that cutoff. I agree with the authors though that older adults are at an increased risk for structural heart disease, systolic dysfunction, valvular heart disease and others, which in turn can manifest with AF (as a presenting disorder), but the alternate is true also. i.e. those who have heart failure for example, can progress with time in terms of their cardiomyopathy and result in AF...This makes the attribution of cost to AF challenging.

Authors’ response: The cut-off age of 50 years has been chosen for two reasons. Reasoning came from the published literature (for example Sankaranarayanan, 2013) suggesting that, “persistent AF under the age of 50 is often associated with identifiable causes like structural heart disease, hyperthyroidism, or alcohol excess”.

More importantly however, we based the cut-off on indication of oral anticoagulants. Most AF patients on our cohort are also on direct oral anticoagulants, and patients who are 50 years or older are likely to be on anticoagulants only because of AF, while patients younger than 50 could be on anticoagulants for reasons other than AF.

3. A strong rationale on why the study population was decided to include patients who are older than 50 is needed. I feel that this cutoff is subjective specifically when looking at reference 11. If a comprehensive examination of cost attributed to AF is sought, all adults admitted with AF should be included. If this is an examination for older adults, who happen to be at the greatest risk for AF (with the least evidence given the systematic exclusion from clinical trials), a cutoff of > 75 years of age was suggested (Circulation. 2016;133:2103–2122. DOI: 10.1161/CIR.0000000000000380)

Authors' response: Please, see the reply for point (2). A stronger rationale for the use of 50+ cut-off is reported on page 4 and 5 of the main document.

4. While AF can certainly be a “precursor” to stroke and cardiovascular disease that is not truly the case every time, specifically if we examine older adults with structural heart disease. For example, patients with mitral valve disease that progress into a severe form may result in a progressive dilation of the left atrial (and subsequent structural changes like fibrosis), and lead to atrial fibrillation. Attributing the cost of such patients to AF, rather to valvular heart disease is actually a major problem. In order to solve this problem, I think a sensitivity analysis to describe the proportion of patients with structural heart disease (at minimum systolic dysfunction, valvular heart disease, etc.). The cost of AF in this setting should be presented in the context of other associated cardiovascular conditions.

Authors' response: Proportions of patients with structural heart diseases are now presented in the manuscript in the Discussion section. From our hospital data, we are not able to establish causation, and we now acknowledge that fact as a limitation.

5. For costing, was that inflation adjusted? Please clarify...The description of costing is great!

Authors' response: The tariffs used for costing account for inflation, therefore further cost adjustment was not needed. This clarification is now included on page 6 of the main document.

6. Page 6: typo (“skewed”) correct pls.

Authors' response: The typo has now been corrected.

7. Can the authors add another model that is conditional on survival (one year/five years)? (similar rationale to having conditioned on having positively incurred cost). Death is the biggest cost saver - I think it would be informative to estimate the cost attributed to AF among patients who are 5-year survivors.

Authors' response: Additional models for patients who survived and those who died within the five years from AF index hospital admission are now presented in the Appendix and results are discussed in the main text.

8. Regarding covariates (i.e. adjustments), how about cardiovascular procedures (at minimum TEE/Cardioversions, holters, event monitors, echocardiograms, angiograms, PCIs, and other cardiovascular procedures?! (all associated with AF). Without adjusting for these, any incurred cost for a patient who was admitted to the hospital AF, and then received an angiogram/PCI will be attributed to AF, rather than coronary artery disease.

Authors' response: These covariates would definitely improve the accuracy of cost estimation. However, these covariates are not available in our routinely collected data of hospital admissions.

9. The discussion is nicely written, but I would expand it to include some of the points discussed above. Limitation sections should include the points above.

Authors' response: This has now been considered and the limitation section has been expanded to address the points raised above.

10. I would like to congratulate the authors on attempting to quantify the cost associated with AF in this heterogenous population. While I understand the complexity of the topic and effort that went into putting this work together (I enjoyed reading it), I feel that my overall concerns lie in clarifying the definition of the study population (why a cutoff of 50 was chosen rather than 65 or 75) and a better delineation of why cost was attributed to AF rather than other CV conditions or procedures. I am hopeful that the authors can address specially if this manuscript is considered for publication in a large platform like BMJ.

VERSION 2 – REVIEW

REVIEWER	Abdulla A Damluji, MD, MPH Johns Hopkins University, Baltimore, Maryland
REVIEW RETURNED	19-Aug-2019

GENERAL COMMENTS	The investigators responded to the reviewers' comments, but I still have major concerns about this paper and the conclusions. It should not proceed as it stands. 1. Age > 50: The aim of this study, as stated by the authors, was "to quantify the inpatient, outpatient, prescribing, and care home costs associated with atrial fibrillation over a five-year period". This statement is inclusive of ALL patients with AF. The aim was not to quantify the cost attributed to AF for patients older than age 50. As such, the conclusion of the study, as stated by the authors, "compressive approach to estimate costs associated with AF" is not valid. It is actually misleading. The authors categorized age, as a binary variable, with a cutoff of 50 years. The authors stated that the age cutoff of 50 was chosen for 2 reasons: (1) Reasoning based on a review article of Sankaranarayanan et al. (2) And "more importantly" based on age cut off on indication for oral anticoagulation. Both reasons are inaccurate and misleading. For number 1 above. The review that the authors cited was published in a low impact journal. It is a review article and not an original research manuscript on findings from large clinical trial or synthesis of evidence, and it is not a guideline document. That review manuscript reflected the opinion of the authors, and that statement was not a well-accepted consensus position by a major cardiovascular society or guideline documents on atrial fibrillation.
--

Further, in Sankaranarayanan et al paper, the claim that this age of 50 cut-off came from the statement “persistent AF under the age of 50 is often associated with identifiable causes like structural heart disease, hypothyroidism, or alcohol excess.”. Using a study population of patients > 50 years of age based on such opinion based reasoning, and then implying that inpatient, outpatient, and prescribing cost is attributed to ALL patients with AF is misleading. Age is a continuous variable. It is should not be dichotomized, unless there is a good reason to focus on facilitating clinical decision making (e.g. CHADS2 and CHADS2VAsc) where age was dichotomized OR categorized (see below) in order to calculate a risk score. However, when we are trying “to quantify the inpatient, outpatient, prescribing, and care home costs associated with atrial fibrillation over a five-year period” dropping a specific patient group without a sound reasoning leads to a biased results.

For number 2 above, i.e. “indication for oral anticoagulation” is also inaccurate and misleading. CHADS2 score for AF using the cut off of age 75 years of age (not 50); the CHADS2VAsc score uses the categorical cut of 65, 65-74, >=75. Age 50 is not a cut off on indication of oral anticoagulation based on clinical practice guidelines.

I think the authors randomly chose the age 50 without any consensus guidelines on management of AF. As it stands, their conclusion and discussion are misleading in a sense that it implies all their findings are attributed to AF rather to a specific patient population with a number of cardiovascular conditions/comorbidities.

Finally, the revised text is not accurate in the Methods section. If most patient in their cohort > 50 years of age are on anticoagulation, that does not mean that age 50 or above is an indication for anticoagulation. In clinical practice guidelines, the trigger to initiate anticoagulation is CHADS2 and CHADS2VAsc which uses a cut off of 65/75 (not age 50). Paragraph no 2 in the Methods should be removed.

2. The cost is not presented in the context of other cardiovascular conditions. For example, if a patient was admitted with heart failure and developed AF during the same hospital admission, the cost will be attributed to AF, but in reality, that is not true. Some, if not the majority, of the cost is attributed to HF. This is the same for all other cardiovascular conditions or procedures.

3. Contradiction in inclusion/exclusion of study population: The authors make the argument that because AF is due to structural heart disease, hypothyroidism, and alcohol excess, they “arbitrarily” chose not to include patients < 50 years of age. However, for older patients > 50, the same is true. Many older patients present with heart failure, structural heart disease, OR receive cardiovascular procedures and concomitantly they get AF. In these cases, the authors attributed the cost to AF, but in the first example (i.e. adults < 50 years of age) they chose not to include them because they though AF is secondary. I feel a better approach is to include everyone AND then do a sensitivity analysis on characteristics of cohort, medical conditions, and procedures received during index admission for AF. As it stands, the results from the analysis are interesting, but the discussion/conclusion are misleading.

4. Because co-variates to adjust for common cardiovascular

	conditions/procedure are not available, the discussion/conclusion of this study is should reflect that there is a major concern cost is not accurately attributed to AF. 5. The tables on survival bias - i.e. regression on those who are alive and those who are dead should be presented in the main manuscript. Both reviewers (1 and 2) independently noticed that this information is important, but the authors chose to present that in the online supplement.
--	--

VERSION 2 – AUTHOR RESPONSE

Reviewer #2

1.Age > 50: The aim of this study, as stated by the authors, was “to quantify the inpatient, outpatient, prescribing, and care home costs associated with atrial fibrillation over a five-year period”. This statement is inclusive of ALL patients with AF.

The aim was not to quantify the cost attributed to AF for patients older than age 50. As such, the conclusion of the study, as stated by the authors, “comprehensive approach to estimate costs associated with AF” is not valid. It is actually misleading.

The authors categorized age, as a binary variable, with a cutoff of 50 years. The authors stated that the age cutoff of 50 was chosen for 2 reasons:

- (1) Reasoning based on a review article of Sankaranarayanan et al.
- (2) And “more importantly” based on age cut off on indication for oral anticoagulation.

Both reasons are inaccurate and misleading.

For number 1 above. The review that the authors cited was published in a low impact journal. It is a review article and not an original research manuscript on findings from large clinical trial or synthesis of evidence, and it is not a guideline document. That review manuscript reflected the opinion of the authors, and that statement was not a well-accepted consensus position by a major cardiovascular society or guideline documents on atrial fibrillation.

Further, in Sankaranarayanan et al paper, the claim that this age of 50 cut-off came from the statement “persistent AF under the age of 50 is often associated with identifiable causes like structural heart disease, hypothyroidism, or alcohol excess.”. Using a study population of patients > 50 years of age based on such opinion based reasoning, and then implying that inpatient, outpatient, and prescribing cost is attributed to ALL patients with AF is misleading. Age is a continuous variable. It is should not be dichotomized, unless there is a good reason to focus on facilitating clinical decision making (e.g. CHADS2 and CHADS2VASc) where age was dichotomized OR categorized (see below) in order to calculate a risk score. However, when we are trying “to quantify the inpatient, outpatient, prescribing, and care home costs associated with atrial fibrillation over a five-year period” dropping a specific patient group without a sound reasoning leads to a biased results.

For number 2 above, i.e. “indication for oral anticoagulation” is also inaccurate and misleading. CHADS2 score for AF using the cut off of age 75 years of age (not 50); the CHADS2VASc score uses the categorical cut of 65, 65-74, >=75. Age 50 is not a cut off on indication of oral anticoagulation based on clinical practice guidelines.

I think the authors randomly chose the age 50 without any consensus guidelines on management of AF. As it stands, their conclusion and discussion are misleading in a sense that it implies all their findings are attributed to AF rather to a specific patient population with a number of cardiovascular conditions/comorbidities.

Finally, the revised text is not accurate in the Methods section. If most patient in their cohort > 50 years of age are on anticoagulation, that does not mean that age 50 or above is an indication for anticoagulation. In clinical practice guidelines, the trigger to initiate anticoagulation is CHADS2 and CHADS2VAsc which uses a cut off of 65/75 (not age 50). Paragraph no 2 in the Methods should be removed.

The incident AF approach here is logical, but as the authors suggest in the Discussion, captures some but not all incident AF patients and would tend to identify “more severe cases.” In clinical practice, plenty of AF patients who have no symptoms or mild symptoms are diagnosed in ambulatory settings. By not capturing such patients, your study likely overestimates the annual costs of caring for AF patients at the patient level, though it may underestimate the total economic burden of the disease (which was not the aim of the study). This should be acknowledged

Authors’ response: The analyses presented in the manuscript are now inclusive of all AF patients.

2. The cost is not presented in the context of other cardiovascular conditions. For example, if a patient was admitted with heart failure and developed AF during the same hospital admission, the cost will be attributed to AF, but in reality, that is not true. Some, if not the majority, of the cost is attributed to HF. This is the same for all other cardiovascular conditions or procedures.

Authors’ response: As stated in the previous revision, it is not possible from the hospital data we have available to establish causation between structural heart disease and AF. While we acknowledge that this is a limitation of our study, we also recognise that this is likely to have a marginal impact on our conclusions, as the global comprehensive approach used in this study include expenditures that are not necessarily related to AF.

3. Contradiction in inclusion/exclusion of study population: The authors make the argument that because AF is due to structural heart disease, hypothyroidism, and alcohol excess, they “arbitrarily” chose not to include patients < 50 years of age. However, for older patients > 50, the same is true. Many older patients present with heart failure, structural heart disease, OR receive cardiovascular procedures and concomitantly they get AF. In these cases, the authors attributed the cost to AF, but in the first example (i.e. adults < 50 years of age) they chose not to include them because they though AF is secondary. I feel a better approach is to include everyone AND then do a sensitivity analysis on characteristics of cohort, medical conditions, and procedures received during index admission for AF. As it stands, the results from the analysis are interesting, but the discussion/conclusion are misleading.

Authors’ response: The analyses presented in the manuscript are now inclusive of all AF patients. Discussion and conclusion have been changed accordingly.

4. Because co-variates to adjust for common cardiovascular conditions/procedure are not available, the discussion/conclusion of this study is should reflect that there is a major concern cost is not accurately attributed to AF.

Authors' response: As responded in the previous correspondence, these covariates would definitely improve the accuracy of cost estimation. However, these covariates are not available in our routinely collected data of hospital admissions. Nevertheless, because AF is coded at discharge, we can be confident that the estimated costs are attributable to AF.

This is now reflected in the discussion/conclusion in page 15.

5. The tables on survival bias - i.e. regression on those who are alive and those who are dead should be presented in the main manuscript. Both reviewers (1 and 2) independently noticed that this information is important, but the authors chose to present that in the online supplement.

Authors' response: The tables on survival bias are now presented in the main text.

VERSION 3 – REVIEW

REVIEWER	Abdulla A Damluji Johns Hopkins University School of Medicine, Baltimore, MD
REVIEW RETURNED	25-Nov-2019

GENERAL COMMENTS	1. On Page 15, the following statement needs revision from: "...as it would indicate whether costs should be attributable to AF or coronary artery disease" To "...as it would indicate whether costs should be attributable to AF or other forms of structural heart disease"... 2. On page 15, this statement is speculative and not based on any data: "Nevertheless, because AF is coded at discharge, we can be confident that the estimated costs are attributable to AF". Please remove. Otherwise, the authors have addressed my comments. Thank you.
---

VERSION 3 – AUTHOR RESPONSE

Reviewer #2

1. On Page 15, the following statement needs revision from: "...as it would indicate whether costs should be attributable to AF or coronary artery disease"

Authors' response: The statement indicated has now been revised.

2. On page 15, this statement is speculative and not based on any data: "Nevertheless, because AF is coded at discharge, we can be confident that the estimated costs are attributable to AF". Please remove.

Authors' response: The statement indicated has now been removed.